# Film Coating of Phosphorylated Mandua Starch on Matrix Tablets for pH-Sensitive Release of Mesalamine

**DOI:** 10.3390/molecules29133208

**Published:** 2024-07-05

**Authors:** Mayank Kumar Malik, Vipin Kumar, Vinoth Kumarasamy, Om Prakash Singh, Mukesh Kumar, Raghav Dixit, Vetriselvan Subramaniyan, Jaspal Singh

**Affiliations:** 1Department of Chemistry, Gurukula Kangri (Deemed to be University), Haridwar 249407, India; comdt.malik@gmail.com (M.K.M.); jaspal@gkv.ac.in (J.S.); 2Department of Pharmaceutical Sciences, Gurukula Kangri (Deemed to be University), Haridwar 249407, India; raghavdixitphd1985@gmail.com; 3Department of Parasitology and Medical Entomology, Faculty of Medicine, Universiti Kebangsaan Malaysia, Jalan Yaacob Latif, Cheras, Kuala Lumpur 56000, Malaysia; 4Department of Kaya Chikitsa, Rishikul Campus, Haridwar, Uttarakhand Ayurved University, Dehradun 248001, India; omprakashsingh@gmail.com; 5Department of Botany and Microbiology, Gurukula Kangri (Deemed to be University), Haridwar 249407, India; mukesh.najibabad@gmail.com; 6Pharmacology Unit, Jeffrey Cheah School of Medicine and Health Sciences, Monash University, Jalan Lagoon Selatan, Bandar Sunway, Selangor Darul Ehsan 47500, Malaysia; 7School of Bioengineering and Biosciences, Lovely Professional University, Phagwara 144001, India

**Keywords:** pH sensitive release, resistant starch, dip coating

## Abstract

Chemically modified mandua starch was successfully synthesized and applied to coat mesalamine-loaded matrix tablets. The coating material was an aqueous dispersion of mandua starch modified by sodium trimetaphosphate and sodium tripolyphosphate. To investigate the colon-targeting release competence, chemically modified mandua starch film-coated mesalamine tablets were produced using the wet granulation method followed by dip coating. The effect of the coating on the colon-targeted release of the resultant delivery system was inspected in healthy human volunteers and rabbits using roentgenography. The results show that drug release was controlled when the coating level was 10% *w*/*w*. The release percentage in the upper gastric phase (pH 1.2, simulated gastric fluid) was less than 6% and reached up to 59.51% *w*/*w* after 14 h in simulated colonic fluid. In addition to in vivo roentgenographic studies in healthy rabbits, human volunteer studies proved the colon targeting efficiency of the formulation. These results clearly demonstrated that chemically modified mandua starch has high effectiveness as a novel aqueous coating material for controlled release or colon targeting.

## 1. Introduction

Controlling the rate of biologically active chemicals delivered from oral formulations has been prioritised in order to enhance their therapeutic value [1]. In addition, the delivery of bioactive compounds to the colon can be beneficial to lower the administrated dose, to improve the bioavailability of the drug for absorption and to minimise the side effects of the anti-inflammatory drug for local treatments [2]. For gastrointestinal tract targeting in the ileo-colonic region, enteric-coated formulations are frequently employed. However, intra- and inter-individual variations in gastrointestinal conditions (fluid pH and transit durations) can occasionally cause enteric coating failure; consequently, proper targeting is difficult. 

A well-accepted concept for regulated release and colon targeting is to develop drug delivery devices that encapsulate active chemicals with polymeric film coatings [3,4]. For safety, economic, and environmental concerns, great emphasis has been placed on the development of aqueous film coatings rather than organic solvents [1,5]. Polymeric film coatings are frequently employed to achieve controlled release of a therapeutic ingredient from a pharmaceutical product as the coated dosage form allows for sustained and accurate release of drug with better repeatability. 

Cellulosic and acrylic polymers are the main types of polymers utilised primarily in pharmaceutical coatings because of good film-forming capabilities, and these polymers can be used to create robust protective coatings. Although there has been a lot of research on these coatings, there may still be issues when they are sprayed on tablet surfaces. The improper coating content may give the dry coating a tendency to split, the tacky nature of the polymeric films may create undesirable agglomeration within the coating procedure, or the coating may interact with the medicine in the tablet core. Additionally, brittleness in the context of ethyl cellulose (EC), a frequently employed coating polymer in manufacturing may inhibit the creation of a coherent film in situations where high flexibility is essential [6,7,8,9]. Hitherto, there is a demand for innovative polymers with superior film-forming properties to address the shortcomings of existing polymers used in coatings.

For use as a coating material in drug delivery devices, starch has received significant attention given its capacity to generate films [10,11]. Yet, given starch’s hydrophilic properties and in order to avert unintended bioactive release in the upper gastrointestinal track (GIT), starches that are employed as materials of coating films must be tailored to have specific characteristics, such as impermeability and insolubility [12,13]. These characteristics arise from hindrance to enzymatic digestion and acidic gastric pH [14]. Due to the low cost, easy availability and non-toxicity of starch, it has been explored by researchers to develop films for colon-specific delivery, but the majority of investigations to date have concentrated on blending commercially polymers with modified starch to efficiently regulate the physicochemical features of the blended film coatings. Very few studies, however, have focused on the preparation of starch-based coating films for effective colon-specific delivery [1,15,16,17]. In our earlier work, cross-linked phosphorylated mandua starch was successfully synthesized and showed strong hindrance to enzymatic digestion [18]. Therefore, chemically modified mandua starch can be applied as a promising colon-targeting delivery carrier material [19,20,21]. The chemical modification process may convert the native starch into a resistant starch, and this form of starch should be resistant towards enzymatic degradation in the gastrointestinal track. The coating of such polymers over the matrix tablets may protect the entrapped drug from the harsh environment of the GIT and regulate its release at the targeted site. In the current study, an aqueous suspension film coating technique is employed to create chemically modified mandua starch film-coated tablets, which are intended to serve as a colon-specific controlled-release device. There were two main components to the approach. The initial phase involved producing the chemically modified mandua starch films, which can be accomplished synchronously by employing a water-based film coating process and suspensions made of chemically modified mandua starch and polyvinyl alcohol (PVA). To confirm the capability of colon-specific controlled release, the second part examined release tests. The release behaviour of chemically modified mandua starch-coated tablets was examined with in vitro tests (i.e., simulating the human GIT) using mesalamine as a model drug. The in vivo effectiveness of the newly developed tablets was also assessed to comprehend the biocompatibility using X-ray imaging analysis of healthy human volunteer and rabbits.

## 2. Results and Discussions

The chemically modified finger millet starch was prepared using a phosphorylation process, and it has been characterized by different techniques for various parameters, including 31P NMR, 13C NMR, 1H NMR, TGA/DTA, powder X-ray diffractometry, FTIR, swelling index, water absorption capacity, oil absorption capacity, moisture content, ash value and digestion resistibility in various simulated digestive fluids, in our previous study [18]. Tablets prepared as AMST (native finger millet (mandua) starch tablet), PST (chemically modified finger millet starch tablet), PSCPST 10 (chemically modified finger millet starch tablet coated with 10% *w*/*w* coating of chemically modified finger millet starch) with stated array of excipients had thickness ranging from 4.01 to 4.12 mm. Drug content was observed to be uniform among various batches of the tablets and ranged from 100.1 ± 1.607 to 100.4 ± 0.343% *w*/*w*. The hardness of uncoated tablets of alkali-isolated mandua starch matrix tablets, uncoated tablets of chemically modified mandua starch and coated tablets was 1.63, 2.26 and 6.4 Kg/cm^2^, respectively. It has been found that all tablets passed the USP (United States Pharmacopoeia) criteria for friability (<1.00% *w*/*w*) and weight variation as shown in Table 1. Friability determination results confirmed the excellent mechanical strength of the tablets.

### 2.1. Effect of Chemically Modified Mandua Starch Film Coating on Cumulative Drug Release (CDR) of Mesalamine from Chemically Modified Mandua Starch Matrix Tablets

The chemically modified mandua starch film coated matrix tablets were designed. Tablets coated with 10% weight gain were selected for dissolution profiles on the basis of our earlier reported results for the Eudragit S-100 coating [22]. Tablets were used to assess the effects of the phosphorylated starch film coating on CDR% from chemically tailored mandua starch matrix tablets. The results plotted in Figure 1 showed the effect of the coating material on the drug release performance of tablet, and it has been found that tablets exhibited coating-dependent performance. In initial 2 h of dissolution, there was only 5.65% *w*/*w* drug release from chemically modified starch coating tablets in the simulated gastric fluid after 10% *w*/*w* level of coating. It reveals that a 10% *w*/*w* level of coating is adequate to protect the release of the drug in the upper gastric phase (simulated gastric fluid; pH 1.2). The level of film-coating thickness purposely escalates the length of the diffusion pathway for the drug [23,24,25]. A reduction in drug release from the coated tablets may be due to modification of hydroxyl groups of native mandua starch during the phosphorylation process. Chemical modification could provide a superior stereospecific barrier effect, which provides a better capability to protect the tablet from attack by the enzymes in the SIF and acid in the SGF [15]. It was found that tablet developed from chemically modified mandua starch was also successful in controlling the release of drug in the gastric phase (SGF, pH 1.2, 2 h). This can be attributed to the fact that the chemical alteration would enhance the content of colonic digestible starch, which would get digested by microflora of the colon [26]. Parmar et al. [27] suggested that the 10% *w*/*w* coating of resistant starch was significant to control the release of the drug in stomach-like conditions. Plasticizers also have a significant role in film formation as it is not possible to form film alone from resistant starch due to the poor mechanical properties of resistant starch [28]. The presence of plasticizer (glycerol) in the coating may have a synergistic effect on the controlled drug release mechanism of modified mandua starch tablets [29]. It has been reported that only 10% *w*/*w* drug release was observed from resistant starch-coated pellets comprising 15% *w*/*w* plasticizer [15]. The findings of the current investigation are further reinforced by other studies conducted elsewhere [15,30,31,32,33].

The effect of the modified starch coating on the cumulative release of mesalamine in SGF, SIF and SCF dissolution media from matrix tablets is presented in Table 2.

### 2.2. Kinetic Analysis of Dissolution Pattern of Mesalamine

As shown in Table 3, the mechanism of drug release was identified by fitting the data obtained from in vitro dissolution into multiple kinetic models and determining the kinetic constant (K), release exponent (n) and regression coefficient values (*R*^2^*)*. The main models for determining the drug release from prolonged release formulations include Higuchi, Korsmeyer–Peppas, first-order equation, zero-order equation and the regression coefficient, and these models were assessed to determine the best model. It has been found that chemically modified mandua starch-coated mesalamine tablets (10% *w*/*w* coating) followed zero-order release kinetics (*R*^2^ = 0.9880) but uncoated tablets of chemically tailored mandua starch and alkali isolated mandua starch followed first-order kinetics. It has been reported by many researchers that the high-level film-coated (12% *w*/*w*) dosage formulation follows zero-order kinetics due to entirely blocked pores and channels [34,35]. The release exponents (n = 0.858) showed that the drug release mechanism (anomalous transport) is governed by diffusion and swelling. Tarvainen et al. [36] also reported similar findings for the starch acetate coating (12% *w*/*w*), which triggered non-Fickian drug release (n = 0.79).

### 2.3. Bioequivalence Studies

It has become essential to be confident that therapeutic efficacy can be attained when switching from branded to generic medications for affordability [37]. Similarity and difference factors are the measurement of the similarity in percentage (%) dissolution between the two curves and the relative error between the two curves, respectively. Two dissolution profiles are considered comparable and bioequivalent if f_1_ is between 0 and 15 and f_2_ is between 50 and 100 [38]. The values of similarity and difference factors estimated from dissolution data for chemically modified mandua starch tablets are summarised in Table 4. It was observed that the Rescigno index (RI) value for release behaviour in SGF, SIF, and SCF was less than 1 for phosphorylated mandua starch film-coated tablets. It was very near to zero, which indicated the likeness of prepared tablets with marketed tablets.

### 2.4. In Vivo Colon Targeting Proficiency of Chemically Modified Finger Millet Starch Tablets

The results of the roentgenography study are represented for optimized chemically modified finger millet starch-coated mesalamine tablet. The roentgenographs of the rabbit with an empty stomach before administration of the tablet clearly showed the non-appearance of the tablet in the GIT of the test animal (Figure 2). The X-ray images of the tablets showed that the tablet maintained its integrity in the gastric environment. Interestingly, the transit of the formulation through the GIT was easily detectable. The roentgenographs of the tablets taken after 120 min showed that tablet reaches the small intestine in an intact form, revealing that the prepared formulations could be site specific (colon). No release of barium sulphate was observed in the small intestine, signifying that no drug was released in the small intestine. The time necessary to reach the colon was ~5–6 h [39]. Further, it has been found that the tablet reaches the colon after 360 min of administration and was slightly swollen. The findings of the study confirmed the in vitro drug release data. Further, the size of the starch tablet was increased after 480 min of administration, a better sign of drug release in the colon. The findings of the study showed good agreement with the study performed by [40]. Patel and Amin [40] concluded that a tablet coated with ethyl cellulose containing hydrophilic material; polyethylene glycol as an inner coating layer; and methyl acrylate, methyl methacrylate, and methacrylic acid copolymer (Eudragit^®^ FS 30D) as outer coating layer remained intact until it reached the colon (5 h).

The ability of the chemically modified mandua starch-coated tablets to deliver mesalamine was also authenticated by performing in vivo studies in healthy human volunteers. The transit time and the behaviour of the tablets in healthy human volunteers were assessed. Based on the X-ray images taken at specific time points, the gastric transit time of the coated tablet was observed to be 2 h in a fasting state (Figure 3b). The transit time for the small intestine was observed to be 6 h (Figure 3d). Further, the tablet maintained its integrity in the gastric pH of the stomach for 2 h. Further, no sign of disintegration was observed in the small intestine. The radiographs taken after 4 h (Figure 3c) of oral administration showed that tablet reached the ileocecal region and remained intact due to the resistance of the resistant starch film coating to the stomach and intestinal fluid. Further, the radiographs taken at 6 h showed that the tablets slowly swelled in the colon (ascending colon). This can be attributed to a possible effect of the development of pores on the film coating due to polymer degradation by colonic bacteria. Interestingly, complete disintegration of the formulation was not observed after 8 h (3e) post administration. The roentgenographic study verified that the designed drug delivery device can deliver the drug to the colon successfully in healthy volunteers.

## 3. Material and Methods

### 3.1. Materials

Finger millet grains (VL Mandua-352) were received from Vivekananda Parvartiya Krishi Anushandhan Sansthan, Uttarakhand. Potassium dihydrogen phosphate, sodium chloride and dipotassium hydrogen phosphate were purchased from Thermo Fisher Scientific, Mumbai, India. Pancreatin (from porcine pancreas) and pepsin (from porcine gastric mucosa) were supplied by CDH. Magnesium stearate and all other reagents and chemicals utilized in the examinations were received from CDH and HiMedia. Double distilled water was used for all experiments.

### 3.2. Methods

#### 3.2.1. Isolation and Modification of Mandua Starch

Mandua starch was extracted as per our earlier reported method [11] using the alkali isolation approach. Further, the isolated starch was modified using sodium trimetaphosphate and sodium tripolyphosphate in a crosslinking phosphorylation mechanism as per our earlier reported procedure in which the mechanism of phosphorylation was presented [18]. Photographic images of the isolation and modification of mandua starch are shown in Figure 4. The proposed structure of distarch monophosphate formed after the phosphorylation process is shown in Figure 5.

#### 3.2.2. Generation and Characterisation of Coated Tablets

##### Preparation of Core Tablets

Mesalamine was used as a model drug in the tablets. Chemically modified mandua starch was used as a release retardant excipient in the tablets. Tablets were formed using the wet granulation method. Briefly, mesalamine was suspended in 2% *w*/*v* native mandua starch and chemically modified mandua millet starch. In the wet granulation process, corn starch was applied as a granulating agent, and the diluents (chemically modified finger millet starch and native finger millet starch) were mixed thoroughly along with the drug. The wet mass was sieved through a no 22 sieve to obtain uniform size granules. The granules were dried in tray dryer (NSW-148; Narang Scientific Works Pvt. Ltd., New Delhi, India) at 70 °C for 10 h until the constant weight of the granules was obtained. Glidant and lubricant were added to the drug-loaded granules. Finally, the granules loaded with drug and thoroughly mixed with lubricant and glidant were compressed using a tablet punching machine (12 Station, CIP Machineries, Model No-360, Ahmedabad, India).

##### Coating Suspension Preparation

Phosphorylated mandua starch was used as a coating dispersion agent to assess the feasibility of the use of this modified starch as a modified release coating. Coating dispersion of the chemically modified starch was prepared as per our earlier adapted method [22] with slight changes. The function and composition of the coating suspension is given in Table 5, and the recipes for the native mandua starch and chemically modified manuda starch tablets containing mesalamine (uncoated and coated with chemically modified mandua starch) are shown in Table 6. A homogeneous suspension of chemically modified finger millet starch was obtained by dispersing it in distilled water (5 mL) at 25 °C under magnetic stirring for 15 min. Plasticiser (Glycerol) was added to the homogenised dispersion of starch, and the mixture stirred for 5 min. Then, the temperature of the homogeneous dispersion was increased to 90 °C. Further, it was constantly stirred for 20 min to gelatinize the chemically modified mandua starch. Polyvinyl alcohol was dissolved in pre-heated water (5 mL, 70 °C) and added to the starch-glycerol mixture to prepare coating dispersion. Finally, the developed coating dispersion was used to coat the tablets.

The coating of modified starches on the respective tablets was performed using the dip-coating method [22]. In this method, the tablets were dipped in the coating dispersion composed of chemically modified mandua starch, glycerol, polyvinyl alcohol and glycerol for 20 s, followed by thorough drying. The dipping of the tablet in coating dispersion and subsequent drying was continued until the desired weight of the coating was achieved. The coating process was evaluated by weighing the coated tablets at different time intervals, and the process was continued until the achievement of 10% *w*/*w* weight gain. The following equation was used to calculate the % coating (weight gain) of the tablets:$$\%\;Coating\;(Weight\;gain)=\frac{w_{2}-w_{1}}{w_{1}}\times 100$$
where W_1_ is the weight (mg) of the uncoated tablet, and W_2_ is the weight (mg) of the coated tablet.

#### 3.2.3. In Vitro Drug Release and Kinetic Evaluation of Dissolution of Mesalamine

The dissolution profiles of coated, uncoated and marketed formulations were investigated using a (USP type II) basket type apparatus. The experimental conditions for the study of mesalamine release from the prepared formulations is shown in Table 7. Simulated gastrointestinal physiological fluids were used to assess the drug release profile characteristics of mesalamine from all the formulations. During the drug dissolution investigation, 5 mL samples were removed from the dissolution basket at specified time periods. An equal quantity of dissolution media preserved at 37 °C was reloaded to dissolution media to conserve the volume of the dissolution media. A syringe-driven membrane filter (HiMedia, 0.22 μm) was used to filter the samples for analysis. The UV-visible spectrophotometer (UV-1800, Shimadzu, Kyoto, Japan) was used for quantitative estimations of mesalamine.

To know the drug release mechanism from the coated and uncoated tablets, the drug release data were fitted into several drug release models, such as the Hixson–Crowell, zero-order, Higuchi, Korsmeyer–Peppas and first-order equations. As the Korsmeyer–Peppas model was specifically considered to confirm the drug release from a polymeric system, dissolution data were fitted to the Korsmeyer–Peppas equation [41].

#### 3.2.4. In Vivo Colon-Specific Proficiency of Chemically Modified Finger Millet Starch Tablets

##### Roentgenographic Study to Assess the Colon Targeting Competency of Tablets Using Barium Sulphate in Rabbits and Healthy Human Volunteers

An ideal investigative tool for tracking oral dosage forms of pharmacologically active components is X-ray radiography. These procedures have numbers of benefits over scintigraphy and magnetic resonance imaging (MRI) because it is simple to access in most research centres or veterinary teaching hospitals; does not necessitate general anaesthesia in animals; can be executed without manual restraint, preventing radiation exposure to staff; and is economical compared with other imaging technologies [42,43]. X-ray radiography also has a myriad of benefits over MRI and scintigraphy [44]. Using barium sulphate as a pharmaceutical carrier, its transition through the gastrointestinal system can be observed visually [45]. The high density and insolubility of barium sulphate in the GIT are two remarkable characteristics [46]. Orally administered tablets containing barium sulphate can be monitored in vivo because it is an inert substance that does not interact with other excipients used in tablet [47]. The aim of the present study was first to scrutinise the colon-targeting competency of the developed tablet using barium sulphate as radio opaque material in X-ray imaging.

(i).Preparation of barium sulphate tablets

The procedure followed to prepare the barium sulphate tablet was similar to that adopted for mesalamine matrix tablet [22,48]. However, the drug was replaced with barium sulphate (radio opaque material). The coating was performed using a procedure similar to that used for the optimized batch.

(ii).Protocols and use of animal

The study was carried out using t standard protocols for the use and care of laboratory animals (CPCSEA guidelines) and was approved by the Institutional Ethics Committee (Approved Protocol number: BMRL/AD/CPCSEA/IAEC/2019/10/02-I). Rabbits were acclimatized to a 12 h dark/light cycle at 25 °C and were given free access to water and food ad libitum. An optimized tablet formulation was administered in the morning on empty stomach. The tablet was orally administered to the animal using a pill dispenser followed by flushing with 15 mL of water. The rabbits were physically active during the investigation period. This protocol was kept constant during the course of the study in order to minimise artefacts. An X-ray machine (Epsilon) was used to record the X-ray images. X-ray imaging of the rabbit abdomen was performed by keeping the rabbits in a prone position. The images were periodically taken at 0 h, 2 h, 4 h, 6 h and 8 h time intervals to investigate the transit behaviour of tablet in the GIT of the rabbit.

(iii).Study protocol for roentgenographic study to assess the colon targeting competence of tablets in healthy human volunteers

To conduct the roentgenography analysis, prior consent from the human ethical committee of Rishikul Campus, Haridwar, Uttarakhand Ayurved University, Dehradun, India, was obtained, and the study was performed under expert guidance after written patient consent (Approval Nos: UAU/RC/IEC/2019/04-02-45 and CTRI/2021/09/036142). Four healthy, male adult non-smokers, non-alcoholic volunteers 20 to 35 years of age were screened for the investigation. All the volunteers were found fit for the study as declared by the physician. All the volunteers abstained from consuming any other medicine for at least one week prior to study initiation. The subjects underwent overnight fasting before the day of test formulation administration [49]. After 12 h fasting, each subject orally swallowed chemically modified mandua starch-coated barium sulphate tablets with sufficient water. X-ray images of the GIT were captured using an X-ray machine (Epsilon) at intervals of 0 h, 2 h, 4 h, 6 h and 8 h to monitor the location, integrity and movement of the tablet in the GIT [50,51].

(iv). Bioequivalence Study of Coated Tablets

The release characteristics of model drug from newly designed tablets of chemically modified starch with a chemically modified mandua starch coating were compared with the marketed tablets using model-independent tactics. Pair-wise procedures, such as Rescigno indices (ξ_1_ and ξ_2_), difference factor (f_1_) and similarity factor (f_2_), were estimated for determination of the release features under the permissible limit [22,41].

#### 3.2.5. Statistical Analysis

The investigations were accomplished in triplicate (n = 3). The *p* value was fixed at 0.05. The statistical treatment of data was performed using one-way ANOVA and student *t*’ test. For statistical analysis, Microsoft Excel 2016^®^ was applied.

## 4. Conclusions

Tailored mandua starch seems to be a promising film former for pharmaceutical coatings. It has been found that tailored mandua starch based aqueous dispersion film can be used for fail safe colonic delivery of the mesalamine. It has been clear form in vitro drug release findings of coated tablets that 10% coating was significant to hinder the drug release in simulated gastric and intestinal fluids. Further, in vivo roentgenographic studies proved colon targeting of the formulation in healthy human volunteer and rabbits. The in vitro drug release and roentgenographic study also demonstrated that the film coating is uniform, and no cracks occurred after drying given the close packing of the branched phosphorylated mandua starch polymeric chains. This in vitro and in vivo characterization of mandua starch film-coated mesalamine tables encourages the development of an aqueous-based chemically modified mandua starch coating formulation for targeting the drug to the colon.

## Figures and Tables

**Figure 1 molecules-29-03208-f001:**
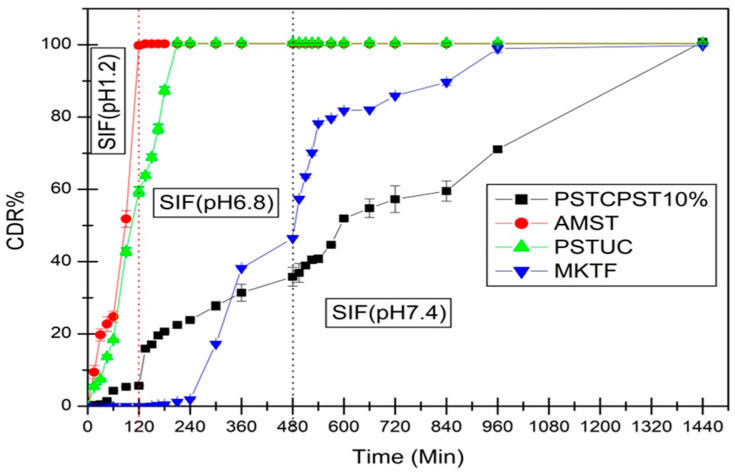
Comparative cumulative drug release (CDR%) of mesalamine in changing over media from native finger millet starch tablet (AMST), chemically modified finger millet starch uncoated tablet (PSTUC), chemically modified finger millet starch tablet with 10% *w*/*w* coating of chemically modified mandua starch (PSTCPST10%), marketed tablet (MKTF, mesalamine: 800 mg) (Ref. [22]).

**Figure 2 molecules-29-03208-f002:**
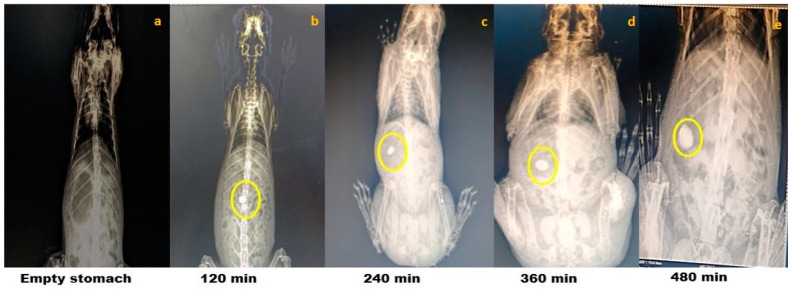
Roentgenography study of optimized phosphorylated mandua starch tablets in rabbits. Gastrointestinal transit of the colon-targeted tablets in rabbits: (**a**) 0 min, no tablet in the stomach; (**b**) 120 min, tablet approaching the small intestine; (**c**) 240 min, tablet approaching the small intestine; (**d**) 360 min, tablet reaches in the colon; and (**e**) 480 min, tablet in the colon.

**Figure 3 molecules-29-03208-f003:**
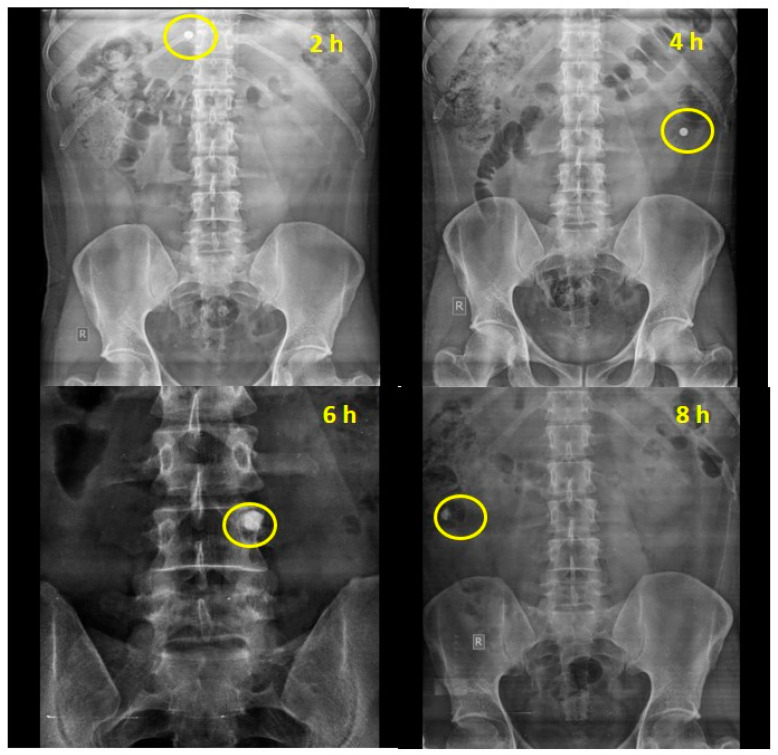
Empty stomach study of chemically modified mandua starch-coated tablet composed of phosphorylated mandua starch.

**Figure 4 molecules-29-03208-f004:**
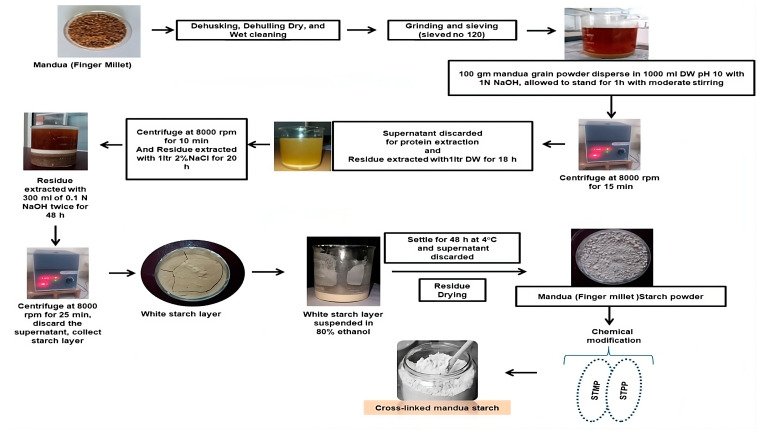
Schematic representation of the procedure for the isolation and modification of mandua starch.

**Figure 5 molecules-29-03208-f005:**
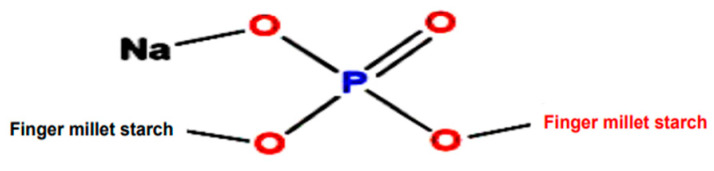
Structure of finger millet starch after the phosphorylation process.

**Table 1 molecules-29-03208-t001:** Physical properties of mesalamine matrix tablets prepared using chemically modified finger millet starch and coated with phosphorylated finger millet starch film.

F. Code	Drug Content (%*w*/*w*) Mean ± SD	Hardness (kg/cm^2^ ± SD)	Weight Variation (mg ± SD)	%Friability (Mean ± SD)	Diameter (mm ± SD	Thickness (mm ± SD)	Disintegration Status in pH 1.2 SGF (1 h)
AMST	100.2 ± 1.25	1.63 ± 0.12	201 ± 2.2	0.55 ± 0.10	8.71 ± 0.01	4.01 ± 0.05	D*
PST	100.4 ± 0.34	2.26 ± 0.09	201 ± 2.2	0.58 ± 0.10	8.71 ± 0.26	4.01 ± 0.22	DP**
PSCPST10	100.1 ± 1.60	6.4 ± 0.54	223 ± 2.7	0.36 ± 0.12	8.92 ± 0.12	4.12 ± 0.30	ND***

AMST: Native finger millet (mandua) starch tablet, PST: chemically modified finger millet starch tablet, PSTCPST10: chemically modified finger millet starch tablet coated with 10% *w*/*w* coating of chemically modified finger millet starch, D*: Disintegrated; ND***: Not disintegrated; DP**: Disintegrated partially, SD: standard deviation of three consecutive determinations. Ref. [22].

**Table 2 molecules-29-03208-t002:** Effect of the modified starch coating (10% *w*/*w*) on the cumulative drug release (%) from chemically modified mandua starch matrix tablets.

F. Code	Cumulative Drug Release (CDR %)		
	SGF (pH 1.2), 2 h	SIF (pH 6.8), 8 h	SCF (pH 7.4), 14 h
PSTCPST10%	5.65	35.80	59.51
PSTUC	59.46	100.5	TD
MKTF	0	57.37	99.45
AMST	99.82	TD	TD

PSTCPST10% = chemically modified finger millet starch tablet with 10% *w*/*w* coating of chemically modified mandua starch, PSTUC = chemically modified finger millet starch uncoated tablet, MKTF = marketed tablet (mesalamine: 800 mg), AMST = native finger millet (mandua) starch tablet, SGF = simulated gastric fluid (pH 1.2), SIF = simulated intestinal fluid (pH 6.8), SCF = simulated intestinal fluid (pH 7.4), TD = totally disintegrated/dissolved

**Table 3 molecules-29-03208-t003:** Release kinetics of mesalamine in media from native finger millet starch tablet (AMST), chemically modified finger millet starch uncoated tablet (PSTUC), chemically modified finger millet starch tablet with 10% *w*/*w* coating of chemically modified finger millet starch (PSTCPS10%) and marketed tablet (MKTF, mesalamine: 800 mg).

F. Code	Zero Order		First Order		Higuchi		Korsmeyer–Peppas			Hixson-Crowell		T_80%_
	*K_o_*	*R^2^*	K_1_	*R^2^*	K_H_	*R^2^*	n	*R^2^*	K_Kp_	K_HC_	*R* ^2^	
PSTCPST10%	0.07	0.98	0.00	0.98	1.93	0.97	0.85	0.99	0.19	0.00	0.98	1052.39
* PSTUC	0.07	0.67	0.00	0.98	6.12	0.95	0.11	0.97	49.59	0.00	0.89	443.08
* MKTF	0.10	0.91	0.00	0.94	3.74	0.98	1.98	0.83	0.00	0.00	0.96	796.43
AMST	0.05	0.54	0.01	0.95	3.63	0.64	0.05	0.96	73.27	0.00	0.77	336.83

* Earlier Reported Study [22].

**Table 4 molecules-29-03208-t004:** Bioequivalence study of chemically modified mandua starch-coated tablets with the marketed formulation.

Parameter	Formulation (PSTCPST10%)
Similarity Factor (f_2_)	32.14
Difference Factor (f_1_)	45.55
Rescigno Index (ξ_1_)	0.1629
Rescigno Index (ξ_2_)	0.1735

PSTCPST10% = chemically modified finger millet starch tablet with 10% *w*/*w* coating of chemically modified finger millet starch.

**Table 5 molecules-29-03208-t005:** Composition of the film coating dispersion.

Ingredients (Function)	Amount (%*w*/*w*)
Chemically modified mandua starch (release retardant and film former)	50
Poly Vinyl alcohol (film former)	30
Glycerol (plasticizer)	10
Water (vehicle)	10

**Table 6 molecules-29-03208-t006:** Recipe for native mandua starch and chemically modified manuda starch tablets containing mesalamine (uncoated and coated with chemically modified mandua starch).

	Formulation Code		
Ingredients (mg)	PST	PSTC10%	AMST
PMS	80	80	0
Mesalamine	50	50	50
AMS	20	20	100
HPMC	q.s.	q.s.	q.s.
Talc	5	5	5
Magnesium stearate	5	5	5
Total weight before coating	200	200	200
Chemically modified starch coating (%*w*/*w*)	0	10	0

PMS = chemically modified (phosphorylated) mandua starch, AMS = alkali-extracted mandua starch, HPMC = hydroxypropyl methyl cellulose, PST = chemically modified starch tablet, PSTC10% = chemically modified starch tablet with 10% *w*/*w* coating of modified finger millet (mandua) starch, AMST = alkali-extracted mandua starch tablet, q.s. = sufficient quantity of HPMC added to make the tablet weigh 200 mg.

**Table 7 molecules-29-03208-t007:** Investigation protocol for the analysis of mesalamine release from all the formulations.

Conditions	Acid Stage (Gastric Phase)	Buffer Stage 1 (Small Intestinal Stage)	Buffer Stage 2 (Colon Phase)
Dissolution media	900 mL SGF	900 mL SIF	900 mL SCF
pH	1.2	6.8	7.4
Duration (h)	2	6	8
Rotation speed (rpm)	100	100	100

SGF = simulated gastric fluid (pH 1.2), SIF = simulated intestinal fluid (pH 6.8), SCF = simulated colonic fluid (pH 7.4).

## Data Availability

Data are contained within the article. Other data is unavailable due to ethical restrictions.

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
