# Peer review of "Film Coating of Phosphorylated Mandua Starch on Matrix Tablets for pH-Sensitive Release of Mesalamine"

_molecules, 2024, doi:10.3390/molecules29133208_

Round 1
Reviewer 1 Report
Comments and Suggestions for Authors
The proposed research is devoted to drug delivery system based on biosourced polymer, namely, starch. In introudction the novelty of the study is not revealed. Manuscript contains inaccuracy with font size (line 146, table 6). Also some typos or confusing units are present, for example, table 7 in last two lines it's unclear why there are letter "ksi", in table 1 everything is in w/w% units in last column, but water in ml). Also I couldn't find ethical statement about experiments with rabbits and with humans, which is extremely important. Overall, I recommend reconsider this article after masor revision.
Comments on the Quality of English LanguageModerate editing of English language required. Some senetences are too long and complicated for understanding
Author Response
Comments and Suggestions for Authors
Comment.1. The proposed research is devoted to drug delivery system based on biosourced polymer, namely, starch. In Introduction, the novelty of the study is not revealed.
Response. In revised manuscript, the novelty of the work has been added.
Comment.2.Manuscript contains inaccuracy with font size (line 146, table 6).
Response. The font size in revised manuscript has been changed and made uniform.
Comment.3. Also some typos or confusing units are present, for example, table 7 in last two lines it's unclear why there are letter "ksi",
Response. The letter ξ indicates Rescigno index in table 7.
Comment.4.In table 1 everything is in w/w% units in last column, but water in ml).
Response. In revised manuscript, the units in table 1 have been made uniform.
Comment.5. Also I couldn't find ethical statement about experiments with rabbits and with humans, which is extremely important. Overall, I recommend reconsider this article after major revision.
Response. The ethical information regarding approval form competent authorities has been added in revised manuscript (Section:2.2.4.1.(ii) for animal study: Approved Protocol number- BMRL/AD/CPCSEA/IAEC/2019/10/02-I and for human trial; section: 2.2.4.1.(iii) with Approved protocol no- UAU/RC/IEC/2019/04-02-45 and CTRI/2021/09/036142)
Comments on the Quality of English Language
Comment.1. Moderate editing of English language required. Some sentences are too long and complicated for understanding
Response. The language of the revised manuscript has been changed wherever necessary.
Note: All the changes has been made in revised manuscript by red colour

Reviewer 2 Report
Comments and Suggestions for Authors
In the original article entitled ”Film coating of phosphorylated mandua starch on matrix tablets for pH sensitive release of mesalamine”, the authors focus on the formulation of new materials used for controlled drug delivery to the ileo-colonic region.
I appreciate the consistency, based on the numerous articles on this topic if we consult the list of bibliographical references (several articles are cited on the topic of this type of starch), in the study and superior valorization of an indigenous raw material, namely mandua starch. The authors propose a chemically modified aqueous-based variant for pharmaceutical formulations with controlled and targeted release of mesalamine in the colon.
Starch has some already known advantages in terms of low cost, lack of toxicity, availability, which is why it has been intensively researched especially in blended coated films, but here, the authors come with the novelty of starch-based coating films for effective colon-specific delivery.
Both in vitro mesalamine release studies and in vivo roentgenography studies in healthy human volunteers and rabbits are interesting, well-conducted and thorough, and have clearly demonstrated targeted and safe colonic release of mesalamine from 10% mandua starch-coated film-coated tablets. This type of film can be used successfully in the pharmaceutical industry for controlled release purposes.
Author Response
Comments and Suggestions for Authors
Comment.1. In the original article entitled ”Film coating of phosphorylated mandua starch on matrix tablets for pH sensitive release of mesalamine”, the authors focus on the formulation of new materials used for controlled drug delivery to the ileo-colonic region.
I appreciate the consistency, based on the numerous articles on this topic if we consult the list of bibliographical references (several articles are cited on the topic of this type of starch), in the study and superior valorization of an indigenous raw material, namely mandua starch. The authors propose a chemically modified aqueous-based variant for pharmaceutical formulations with controlled and targeted release of mesalamine in the colon.
Starch has some already known advantages in terms of low cost, lack of toxicity, availability, which is why it has been intensively researched especially in blended coated films, but here, the authors come with the novelty of starch-based coating films for effective colon-specific delivery.
Both in vitro mesalamine release studies and in vivo roentgenography studies in healthy human volunteers and rabbits are interesting, well-conducted and thorough, and have clearly demonstrated targeted and safe colonic release of mesalamine from 10% mandua starch-coated film-coated tablets. This type of film can be used successfully in the pharmaceutical industry for controlled release purposes.
Response.1. Thank you very much for the comments of the reviewer regarding the quality and novelty of the manuscript as well as recommendation for publication.

Reviewer 3 Report
Comments and Suggestions for Authors
The authors studied Film coating of phosphorylated mandua starch on matrix tablets for pH sensitive release of mesalamine.
Lines 47 and 48: The authors should represent citations for the sentence.
Line 69: What does GIT stand for? The authors should not have used abbreviations for the usage of the first time.
Line 76 the sentence indicated that there are a few studies but the authors did not represent any citations.
Line 79 the reference 2023 a or b?
Figure 1 is a very similar figure used in ‘Malik, M. K., Kumar, V., Singh, J., Bhatt, P., Dixit, R., & Kumar, S. (2023). Phosphorylation of Alkali Extracted Mandua Starch by STPP/STMP for Improving Digestion Resistibility. ACS omega, 8(13), 11750-11767.’ The figures are slightly different than each other, and the photos were not represented in the 2023b citation. Please indicate it in the text.
Line 125 the citation belong to a or b?
Line 139: What are the ingredients for coating solution?
Line 146: waight should be weight.
Table 1: It is not clear whether the weights are in grams or mg. And the composition is not clearly represented.
Table 2: what do PST, PSTC10%, AMST, AMS, HPMS, and Talc stand for? What does q.s stand for? What * indicates?
Table 3. What do SGF, SIF, and SCF stand for?
Line 172-177: The sentences need citations.
Line 179-182: The sentence needs citations.
The aim is one, please do not use the plural form.
Line 185-189: The authors claim to follow the protocol, what is the protocol and the citation?
Line 200-202: How did the Authors define the time intervals? Any citations?
Line 211-216: the sentences need citations.
Line 225-227: The software used for the statistical analysis should be represented.
Line 229: What do AMSR, PST and PCPST 10 stand for?
Line 242: is the reference a or b?
Line 247: is the reference a or b?
Line 256-257: Are there any references found with the same results?
Line 265-268: The sentences need citations.
Figure 2: What does CDR stand for? Are AMST and PSTUC represented in Malik et. Al., 2023 b study?
Table 5: What do PST-PS, PST, MKTF, AMST stand for? What is TD? What * indicates?
Line 280-281: The sentence should be represented before the table.
Line 302: the reference a or b?
Line 329: The sentence needs citation.
Line 334: The citation is in the wrong place.
Figures 3 and 4: These are 2D images and are represented nicely but did the authors check the feces for the tablet residues? Did the authors study the dimensions of the tablets? Is there any proof that shows the tablets release their content in the colon?
The references are limited and should be renewed with the more recent references.
Author Response
Comments and Suggestions for Authors
The authors studied Film coating of phosphorylated mandua starch on matrix tablets for pH sensitive release of mesalamine.
Comment.1.Lines 47 and 48: The authors should represent citations for the sentence.
Response.1. Thank you for the suggestion. The citations of the sentence have been added in the revised manuscript.
Comment.2.Line 69: What does GIT stand for? The authors should not have used abbreviations for the usage of the first time.
Response.2. The full form of abbreviation ‘GIT’ has been added in the revised manuscript.
Comment.3. Line 76 the sentence indicated that there are a few studies but the authors did not represent any citations.
Response.3. The concerned citations have been added in revised manuscript (Chen et al., 2018; Pu et al., 2011; Chen et al., 2005; Khan et al., 2020).
Comment.4.Line 79 the reference 2023 a or b?
Response.4. Thank you for pointing out the missing information. The concerned reference is Malik et al.2023a.
Comment.5.Figure 1 is a very similar figure used in ‘Malik, M. K., Kumar, V., Singh, J., Bhatt, P., Dixit, R., & Kumar, S. (2023). Phosphorylation of Alkali Extracted Mandua Starch by STPP/STMP for Improving Digestion Resistibility. ACS omega, 8(13), 11750-11767.’ The figures are slightly different than each other, and the photos were not represented in the 2023b citation. Please indicate it in the text.
Response.5. In figure 1 of the manuscript, both the process of alkali assisted extraction of starch and the phosphorylation process have been shown. The alkali extraction process of starch and the phosphorylation process was adopted as per reference Malik et al. 2023a (Malik, M. K., Kumar, V., Singh, J., Bhatt, P., Dixit, R., & Kumar, S. (2023). Phosphorylation of Alkali Extracted Mandua Starch by STPP/STMP for Improving Digestion Resistibility. ACS omega, 8(13), 11750-11767). The required text has been added in revised manuscript.
Comment.6.Line 125 the citation belong to a or b?
Response.6. The citation belongs to 2023b and it has been corrected in revised manuscript.
Comment.7. Line 139: What are the ingredients for coating solution?
Response.7. The ingredients of coating solution has been incorporated in revised manuscript.
Comment.8. Line 146: waight should be weight.
Response.8. The typographical error has been corrected and it has been written as weight
Comment.9.Table 1: It is not clear whether the weights are in grams or mg. And the composition is not clearly represented.
Response.9. The composition of ingredients of coating has been clarified in table.1.
Comment.10. Table 2: what do PST, PSTC10%, AMST, AMS, HPMC, and Talc stand for? What does q.s stand for? What * indicates?
Rsponse.10. The full names of all abbreviations have been added with Table.2.
Comment.11. Table 3. What do SGF, SIF, and SCF stand for?
Response.11. The full forms of abbreviations e.g. SGF, SIF, and SCF have been added with Table.3.
Comment.12.Line 172-177: The sentences need citations.
Response.12. The citations have been added in revised manuscript.
Comment.13. Line 179-182: The sentence needs citations.
Response.13. The citations have been added in revised manuscript.
Comment.14.The aim is one, please do not use the plural form.
Response. 14. The typographical error has been corrected in revised manuscript.
Comment.15. Line 185-189: The authors claim to follow the protocol, what is the protocol and the citation?
Response.15. The citation has been added related in revised manuscript. The protocol is related to development of tablets with and without barium sulphate.
Comment.16. Line 200-202: How did the Authors define the time intervals? Any citations?
Response.16. The citation has been added. The time interval was decided on the basis of transition time of formulation from gastric to intestinal regions. Within 2 h, the tablet should remain in stomach and at 4 and 8 h, the tablet should transit from stomach to small intestine. At 8h, the tablet should be present in large intestine.
Comment.17. Line 211-216: the sentences need citations.
Response.17. The citations have been added.
Comment.18. Line 225-227: The software used for the statistical analysis should be represented.
Response. 18. For statistical analysis, Microsoft excel 2016 was applied and it has been added in the revised manuscript.
Comment.19. Line 229: What do AMSR, PST and PCPST 10 stand for?
Response. 19. The full forms of these abbreviations have been added in revised manuscript.
Comment. 20. Line 242: is the reference a or b?
Response.20. The reference is 2023b and it has been changed accordingly in revised manuscript.
Comment.21. Line 247: is the reference a or b?
Response.21. The reference is 2023b and it has been changed accordingly in revised manuscript.
Comment.22.Line 256-257: Are there any references found with the same results?
Response.22. The references related to the similar results have been added in revised manuscript.
Comment.23. Line 265-268: The sentences need citations.
Reference. 23. The citations have been added with the required sentences.
Comment.24. Figure 2: What does CDR stand for? Are AMST and PSTUC represented in Malik et. Al., 2023 b study?
Response.24. In figure 2, CDR stands for cumulative drug release and it has been added in revised manuscript. In Malik et. al., 2023 b, native finger millet starch tablet (AMST) and chemically modified (phosphorylated) finger millet starch uncoated tablets were also used as reference.
Comment.25. Table 5: What do PST-PS, PST, MKTF, AMST stand for? What is TD? What * indicates?
Response. 25. The full forms of all abbreviations have been added with table.5.
Comment.26.Line 280-281: The sentence should be represented before the table.
Response.26.The sentence has been added before the table in revised manuscript.
Comment. 27. Line 302: the reference a or b?
Response. 27. The reference is Malik et al.2023 b and has been added in revised manuscript.
Comment.28.Line 329: The sentence needs citation.
Response. 28. The citation has been added in the revised manuscript.
Comment.29. Line 334: The citation is in the wrong place.
Response.29. The citation has been corrected in revised manuscript.
Comment. 30. Figures 3 and 4: These are 2D images and are represented nicely but did the authors check the feces for the tablet residues? Did the authors study the dimensions of the tablets? Is there any proof that shows the tablets release their content in the colon?
Response.30. In this study, the feces of human volunteers and rabbits was not analyzed for the residues of the tablet. But in imaging, the surface of the tablet in images of human volunteers acquired at 6 and 8 h was not smooth as observed in the image acquired at 2h and 4h interval. The rough and irregular surface indicated the effect of biological fluids on tablets. In the image acquired at 8h, the size of tablet was comparatively less and the surface was also irregular. As the size of the tablet was reduced and it would be due the impact of colonic fluids on tablets. Due to degradation of tablet, the size would be reduced. As the tablet was remained in colonic region upto evacuation, the contents would be degraded during this retention time in colon.
Comment.31. The references are limited and should be renewed with the more recent references.
Response. 31. As per suggestions, more recent references have been added in revised manuscript.

Reviewer 4 Report
Comments and Suggestions for Authors
In the investigation conducted by M. K. Malik et al., a study was presented on newly designed modified starch matrix tablets aimed at enabling pH-sensitive release of the drug mesalamine. The authors employed previously modified starch from their earlier study, utilizing it in the formulation of drug-containing tablets, which were further coated with modified starch using a specific dip-coating process. The study involved an evaluation of in-vivo drug release kinetics, with further investigations conducted on both animal models and human volunteers. The results provided by the authors indicated that the designed controlled drug release system was activated at colon pH, demonstrating promising properties for targeted delivery of mesalamine to colon tissues.
The study was effectively presented, and the results provided substantial support for the primary objective. However, before considering publication in 'Molecules,' there are several points that the authors should address:
1. Line 28: "Reontogenography" should be modified to "roentgenography."
2. In Method section 2.2, it would be more beneficial if the authors provided the chemical structure of the modified starch within the manuscript to assist readers in following the study.
3. The previously characterized modified starch was presented in a study by the authors (10.1021/acsomega.2c05783). However, it would be more appropriate for the authors to provide a statement indicating the successful characterization of the synthesized polymer, including FTIR, TGA, and other relevant analyses.
4. Line 130: "Distilled water" is mistakenly duplicated.
5. Table 2: The authors should clarify the meaning of PST, PSTC, and AMST and consider removing these designations from Table 4.
6. Table 3: Clarify the meaning of SGF and SIF.
7. Line 234: Provide "United States Pharmacopeia" after USP within parentheses.
8. Figure 3: The representative roentgenography study images should be presented in the same size for comparative analysis.
9. Almost 30% of the citations provided belong to the authors' previous studies. It would be more appropriate for the authors to reference more recent studies rather than relying heavily on self-citations.
Author Response
Comments and Suggestions for Authors
In the investigation conducted by M. K. Malik et al., a study was presented on newly designed modified starch matrix tablets aimed at enabling pH-sensitive release of the drug mesalamine. The authors employed previously modified starch from their earlier study, utilizing it in the formulation of drug-containing tablets, which were further coated with modified starch using a specific dip-coating process. The study involved an evaluation of in-vivo drug release kinetics, with further investigations conducted on both animal models and human volunteers. The results provided by the authors indicated that the designed controlled drug release system was activated at colon pH, demonstrating promising properties for targeted delivery of mesalamine to colon tissues.
The study was effectively presented, and the results provided substantial support for the primary objective. However, before considering publication in 'Molecules,' there are several points that the authors should address:
Comment.1. Line 28: "Reontogenography" should be modified to "roentgenography."
Response.1. The required correction has been made in revised manuscript.
Comment.2. In Method section 2.2, it would be more beneficial if the authors provided the chemical structure of the modified starch within the manuscript to assist readers in following the study.
Response.2. The chemical structure of chemically modified finger millet starch has been added in revised manuscript.
Comment.3.The previously characterized modified starch was presented in a study by the authors (10.1021/acsomega.2c05783). However, it would be more appropriate for the authors to provide a statement indicating the successful characterization of the synthesized polymer, including FTIR, TGA, and other relevant analyses.
Response.3. As per suggestions, the characterization of chemically modified starch has been added in revised manuscript with citation in results section.
Comment.4. Line 130: "Distilled water" is mistakenly duplicated.
Response.4. The typographical error has been rectified in revised manuscript.
Comment.5.Table 2: The authors should clarify the meaning of PST, PSTC, and AMST and consider removing these designations from Table 3.
Response.5. The full forms of abbreviations have been added with table. 3 as well as table 4 in revised manuscript.
Comment.6. Table 3: Clarify the meaning of SGF and SIF.
Response.6. The full forms of abbreviations have been added with table. 3
Comment.7.Line 234: Provide "United States Pharmacopeia" after USP within parentheses.
Response.7. Within parenthesis, the full form of USP has been added.
Comment.8. Figure 3: The representative roentgenography study images should be presented in the same size for comparative analysis.
Response.7. Thank you for the suggestion. Figure 3 (revised Figure 4) has been replaced with new image with similar size.
Comment.9. Almost 30% of the citations provided belong to the authors' previous studies. It would be more appropriate for the authors to reference more recent studies rather than relying heavily on self-citations.
Response.9. More recent citations have been added in revised manuscript.

Reviewer 5 Report
Comments and Suggestions for Authors
Comment 1: “2.2.2. (i). Preparation of Core Tablets” The procedure is described in too little detail and cannot be reproduced.
Comment 2: “The homogeneous suspension of chemically modified finger millet starch was obtained by dispersing in distilled water was dispersed in distilled water” ??? The sentence is illogical.
Comment 3: The font size often changes in the text. Check it.
Comment 4: Table 5: What does TD mean?
Comment 5: Release occurred at several pHs. Do kinetic models make sense in this case?
Comment 6: Figure 1: Please improve the quality.
Author Response
Comments and Suggestions for Authors
Comment 1: “2.2.2. (i). Preparation of Core Tablets” The procedure is described in too little detail and cannot be reproduced.
Response.1. The procedure for core tablet formation has been elaborated in revised manuscript.
Comment 2: “The homogeneous suspension of chemically modified finger millet starch was obtained by dispersing in distilled water was dispersed in distilled water” ??? The sentence is illogical.
Response.2. The above sentence has been corrected in revised manuscript.
Comment 3: The font size often changes in the text. Check it.
Response.3. The font size throughout the manuscript has been checked and changed to uniform size.
Comment 4: Table 5: What does TD mean?
Response. In table 5, TD stands for totally dissolved and the full form of the abbreviation has been added in the revised manuscript.
Comment 5: Release occurred at several pHs. Do kinetic models make sense in this case?
Response. 5. The release performance of mesalamine from coated tablets was assessed in simulated gastric fluid (pH 1.2), simulated intestinal fluid (pH 6.8) and simulated colonic fluid (pH 7.4). As the drug was entrapped in coated tablet, the release would be affected by the nature of film as well of the matrix. In acidic and basic pHs used for dissolution study, the release was governed by diffusion as well as erosion of the polymeric matrix. The diffusion of drug along with erosion/degradation of polymers would certainly be affected by dissolution media. In acidic pH, the drug release was comparatively less and it would be due to diffusion. In basic pH, the more release of drug would be due to erosion, degradation, disintegration along with diffusion mechanism.
Comment 6: Figure 1: Please improve the quality.
Response.6. The quality of the figure 1 has been improved in revised manuscript (dpi 1200).

Round 2
Reviewer 1 Report
Comments and Suggestions for Authors
Authors addressed all listed issues, especially about ethical requirements. I recommend to accept paper in present form
Comments on the Quality of English LanguageMinor English editing is required
Author Response
Reviewer 1
Comments and Suggestions for Authors
Comment.1. Authors addressed all listed issues, especially about ethical requirements. I recommend to accept paper in present form
Response.1. Thank you for your comment and recommending the manuscript for the publication.
Comments on the Quality of English Language
Comment.2. Minor English editing is required
Response.2. The editing of English has been done wherever required in the revised manuscript.
Note: All the changes have been shown by underline text.

Reviewer 3 Report
Comments and Suggestions for Authors
The Authors corrected the suggestions.
Author Response
Reviewer 3
Comments and Suggestions for Authors
Comment.1. The Authors corrected the suggestions.
Response.1. Thank you for your comment
